# Chronic Lymphocytic Leukemia (CLL)-Derived Extracellular Vesicles Educate Endothelial Cells to Become IL-6-Producing, CLL-Supportive Cells

**DOI:** 10.3390/biomedicines12071381

**Published:** 2024-06-21

**Authors:** Orit Uziel, Lian Lipshtein, Zinab Sarsor, Einat Beery, Shaked Bogen, Meir Lahav, Alon Regev, Vitali Kliminski, Roded Sharan, Asia Gervits, Lorenzo Federico Signorini, Shai Shimony, Pia Raanani, Uri Rozovski

**Affiliations:** 1The Felsenstein Medical Research Center, Rabin Medical Center Petah-Tikva, Petah Tikva 49100, Israel; lian28886@gmail.com (L.L.); zinab.amer.1993@gmail.com (Z.S.); einatb@clalit.org.il (E.B.); mlahav@tauex.tau.ac.il (M.L.); alonreg27@gmail.com (A.R.); vitalikl@tauex.tau.ac.il (V.K.); shaishimony@gmail.com (S.S.); piar@clalit.org.il (P.R.); rozovski.uri@gmail.com (U.R.); 2Institute of Hematology, Davidoff Cancer Center, Petah Tikva 49100, Israel; shakednoah@gmail.com; 3Faculty of Medicine, Tel-Aviv University, Ramat-Aviv, Tel Aviv 6997801, Israel; 4Blavatnik School of Computer Science, Tel-Aviv University, Ramat-Aviv, Tel Aviv 69978, Israel; roded@tauex.tau.ac.il (R.S.); asiagervits@tauex.tau.ac.il (A.G.); lorenzos@tauex.tau.ac.il (L.F.S.)

**Keywords:** CLL, EVs, endothelial cells

## Abstract

We hypothesized that via extracellular vesicles (EVs), chronic lymphocytic leukemia (CLL) cells turn endothelial cells into CLL-supportive cells. To test this, we treated vein-derived (HUVECs) and artery-derived (HAOECs) endothelial cells with EVs isolated from the peripheral blood of 45 treatment-naïve patients. Endothelial cells took up CLL-EVs in a dose- and time-dependent manner. To test whether CLL-EVs turn endothelial cells into IL-6-producing cells, we exposed them to CLL-EVs and found a 50% increase in IL-6 levels. Subsequently, we filtered out the endothelial cells and added CLL cells to this IL-6-enriched medium. After 15 min, STAT3 became phosphorylated, and there was a 40% decrease in apoptosis rate, indicating that IL-6 activated the STAT3-dependent anti-apoptotic pathway. Phospho-proteomics analysis of CLL-EV-exposed endothelial cells revealed 23 phospho-proteins that were upregulated, and network analysis unraveled the central role of phospho-β-catenin. We transfected HUVECs with a β-catenin-containing plasmid and found by ELISA a 30% increase in the levels of IL-6 in the culture medium. By chromatin immunoprecipitation assay, we observed an increased binding of three transcription factors to the IL-6 promoter. Importantly, patients with CLL possess significantly higher levels of peripheral blood IL-6 compared to normal individuals, suggesting that the inducers of endothelial IL-6 are the neoplastic EVs derived from the CLL cells versus those of healthy people. Taken together, we found that CLL cells communicate with endothelial cells through EVs that they release. Once they are taken up by endothelial cells, they turn them into IL-6-producing cells.

## 1. Introduction

Chronic lymphocytic leukemia (CLL) is characterized by a gradual accumulation of mature-appearing, long-lived lymphocytes that travel in the blood and reside in lymph nodes, spleen, and bone marrow [1]. In these sites, pro-inflammatory humoral factors support the survival and proliferation of the neoplastic cells [2]. They do so by inducing intracellular pro-survival pathways such as the B-cell receptor [3], NF-kB [4], or STAT3 [5] in CLL cells. These factors include cytokines and chemokines such as CCL2 [6], CCL3 [7], and IL-10 [7]. Likewise, previous studies have consistently shown that levels of the proinflammatory cytokine IL-6 are at least 10-fold higher in patients with CLL compared with healthy individuals [8]. Yet, which cells produce and secrete IL-6 and what triggers this cellular activity in CLL is unknown.

CLL cells are not simply the seeds that grow on the supportive soil of the microenvironment but play an active role in shaping their surroundings [9]. For example, CLL cells aberrantly produce and secrete the T-cell-derived chemokines CCL3 and CCL4. These chemokines attract monocytes, which in lymph nodes become “nurse-like cells” that support CLL cells’ survival [10]. In addition to monocytes, other cellular elements that are present in the microenvironment, such as mesenchymal stromal cells or endothelial cells, are putative pro-survival agents that the neoplastic cells recruit. 

Secreted by all types of cells, EVs are nano-scaled particles that travel in the blood and carry a cargo that at least partially reflects the molecular makeup of its cell of origin [11]. EVs, including those originating from neoplastic cells, function as stable intercellular transport vehicles that deliver their cargo to cells that engulf them [12]. For example, CLL-derived EVs are taken up by mesenchymal stromal cells, transforming them into cancer-associated fibroblasts [13]. 

Given the appropriate stimulation, endothelial cells produce IL-6, which provides CLL cells with a survival advantage [14]. Therefore, we hypothesized that CLL-EVs turn endothelial cells into “IL-6-secreting cells”.

## 2. Materials and Methods

### 2.1. Patients’ Characteristics

Peripheral blood samples were obtained from 45 treatment-naïve patients with CLL who were followed at Rabin Medical Center after they agreed to participate and signed an informed consent. The study received the approval of our Institutional Review Board (072-17-RMC). 

### 2.2. Cell Lines

To capture a wide range of the vascular bed in which EVs come in close proximity to endothelial cells, we used both vein-derived and artery-derived cell lines. 

Human umbilical vein endothelial cells (HUVECs) were purchased from PromoCell, (Heidelberg, Germany). 

Human endothelial aortic cells (HEAOCs) were purchased from CellBiologics Inc. (Chicago, IL, USA). 

The cells were cultured in EV-free growth medium supplemented with 2% low serum growth supplements, 1% penicillin, and streptomycin (PromoCell). 

### 2.3. Isolation of EVs 

An amount of 10 mL of peripheral blood cells was fractionated using Ficoll Hypaque 1077 (StemCell Technologies, Vancouver, BC, Canada). More than 95% of the peripheral blood lymphocytes obtained from these patients were CD19+/CD5+, as assessed by flow cytometry (Gallios, Beckman Coulter, Brea, CA, USA). We then seeded 200–400 × 10^6^ CLL cells and suspended the cells in 5 ml RPMI 1640 EV-free, heat-inactivated medium (Biological-Industries, Beit-Haemek, Israel) and, after 72 h, collected the medium by 10 min centrifuge at 1000 relative centrifugal force (RCF). The EVs were extracted from this medium by serial ultracentrifugation (Optima XE-90 Ultracentrifuge, Beckman Coulter, Brea, CA, USA), as previously described [10]. To purify the ultracentrifuge-derived EV extract, we used magnetic beads conjugated to an anti-human CD81 antibody (Miltenyi Biotech, Rhineland, Germany). After this step, more than 90% of the bound vesicles were CD81-positive. As indirect evidence of possible control, we rely on the finding of higher levels of IL-6 in patients with CLL compared to their healthy individual counterparts [15], assuming that the difference between CLL patients and healthy people is the characteristics of B cells, neoplastic ones in the case of CLL patients versus normal B cells in the healthy population. 

### 2.4. Nanosight Tracking Analysis of EVs

EVs were quantified by Nanosight tracking analysis (NTA), according to the manufacturer’s instructions (Malvren Panalytical, Cambridge, UK). Briefly, samples were diluted 1:100 in particle-free PBS (Biological Industries, Beit Haemek, Israel) and analyzed under constant flow conditions at 25 °C.

### 2.5. Flow Cytometry Analysis

Since CD81 is ubiquitously expressed on all types of EVs [11] and CD19 is expressed by CLL cells and hence CLL-EVs, we double-stained the particle extract with anti-CD81 and anti-CD19 (both purchased from Miltenyi). To isolate the EVs, we used anti-CD81 magnetic beads diluted to 1:10 in PBS and conjugated to phycoerythrin (Milteniey) for 45 min (Beckman). 

### 2.6. Electron Microscopy Imaging 

We loaded 3 μL of EV extract on glow-discharged lacey grids (EmiTech K100 labtech, Sussex, UK) that were blotted and plunged into liquid ethane using a Gatan CP3 automated plunger. The grids were subsequently stored in liquid nitrogen until use. Frozen specimens (samples with EVs embedded in vitreous ice) were transferred to a Gatan 914 cryo-holder and maintained at temperatures below −176 °C inside the microscope. We inspected the samples with a Tecnai G2 microscope (Thermo Fisher Scientific, Waltham, MA, USA). This instrument has an acceleration voltage of 120 kV and is equipped with a cryobox decontaminator. Images were taken using a Digital Micrograph with a Multiscan Camera model 794 (Gatan, Pleasanton, CA, USA). 

### 2.7. Western Blot Assessing EV Markers

Verification of the extracted EVs was determined by Western blotting using antibodies against well-characterized EV protein markers: CD63, CD9, CD81, and TSG101. Extracted EVs were resuspended in RIPA buffer, sonicated, and quantified using Pierce BCA Protein Assay Kit (Thermo Fisher Scientific, Waltham, MA, USA). An amount of 50 μg of protein was subjected to 10% sodium dodecyl sulfate–polyacrylamide gel electrophoresis (SDS-PAGE) and transferred to a nitrocellulose membrane. The membrane was then hybridized for 16 h at 4 °C with antibodies against EV markers: anti-CD63, anti-CD9, anti-CD81 (1:1000, Santa Cruz Biotech, Dallas, TX, USA), and anti-TSG101 (1:500, Abcam, Waltham, MA, USA). On the following day, the membrane was subjected to fluorescent-labeled secondary antibodies. Visualization was performed using the Odyssey analysis software 6.0 (Odyssey IR imaging system; LI-COR, Lincoln, NE, USA). 

### 2.8. EV Uptake

CLL-derived EVs were double stained. First, they were labeled by ED11, a green-staining construct that was generously provided to us by Dr. Dan Offen (Tel Aviv University, Israel). This construct is based on a membranal penetration signal conjugated to Fluorescein isothiocyanate (FITC). Briefly, 200 μL of EVs (10^10^ EVs/mL) was incubated with 100 μgrams of ED11 for 1 h at room temperature. Thereafter, labeled EVs were washed by ultracentrifugation at 100,000 RCF for 2 h. Subsequently, the EVs were incubated with the red FM-1-43 membrane dye (Thermo Fisher Science, Waltham, MA, USA). For the uptake experiments, 50 × 10^3^ HUVECs were seeded in a 6-well plate. After 24 h, 4.5 × 10^8^ of labeled EVs were added. Prior to harvesting the cells, they were washed twice with PBS and fixed in 4% formaldehyde for 10 min at room temperature in the dark. After an additional wash with PBS, a mounting gel containing the nuclear stain 2-[4-(Aminoiminomethyl) phenyl]-1H-Indole-6-carboximidamide hydrochloride (DAPI) was added on top of each slide. The cells were visualized by fluorescent microscopy (×63) (Axiolmager Z1 with Apotome, Oberkochen, Germany) with the following filters: FM-143 (528/617), FITC (490/525), and DAPI (358/461); and by confocal microscopy (Leica Tcs SP8, Leica Microsystems, Vetzlar, Germany) microscopy) (×100) with the following filters: FM-143 (561/565-685) and DAPI (405/450-410).

### 2.9. Preparation of Protein Extracts

Cells were washed twice with 5 mL of cold PBS. Then, 700 μL of lysis buffer was added. The lysis buffer is made at our lab and contains 8 M urea; 40 mM Tris/HCl (pH 7.6); protease inhibitor cocktail (Sigma, Rehovot, Israel); and phosphatase inhibitors (2.5 mM of sodium orthovanadate, 1 mM of NaF). After 10 min of incubation on ice, the lysates were collected and centrifuged at 16,000 RCF at 4 °C for 30 min and kept at −20 °C. We used Nano-Drop ND-1000 (Thermo Fisher Scientific, Waltham, MA, USA) to measure protein concentration. 

### 2.10. Immunoprecipitation

An amount of 40 μL of protein A-G Sepharose beads was washed twice with 500 μL lysis buffer at 400 RCF for 1 min. The beads were immersed in 1 mL of lysis buffer, and antibodies directed at WBP11, CD11A, and β-catenin (all from Abcam, Cambridge, UK) were added according to the recommended concentrations. This mixture was incubated for 4 h at 4 °C with agitation. Then, the beads were washed twice with 500 μL lysis buffer at 400 RCF for 1 min. An amount of 800 μL lysis buffer containing proteasome and phosphatase inhibitors (PI, DTT, PMSF, NAF, and SOV) and 1 mg of HUVEC protein extracts were added and incubated overnight at 40 C with agitation. To prepare the pellet for Western blot, the tubes were centrifuged at 400 RCF for 1 min and washed three times with 500 μL lysis buffer at 400 RCF for 1 min. Each pellet was immersed with 24 μL lysis buffer and with 6 μL sample buffer boiled for 3 min.

### 2.11. Western Immunoblotting

An amount of 50 µg of protein extract was subjected to SDS-PAGE and transferred to a nitrocellulose membrane. The membrane was hybridized overnight at 4 °C (with agitation) with anti-serine pWBP11, anti-serine pCD11A, anti-serine p β-catenin, and anti-tyrosine pSTAT3 (all from Abcam, Cambridge, UK). On the following day, the membrane was subjected to fluorescently labeled anti-mouse or anti-rabbit antibodies (LI-COR Biosciences, Lincoln, NE, USA). Visualization and densitometry analysis were performed using Odyssey analysis software (Odyssey IR imaging system; LI-COR Biosciences).

### 2.12. Phosphoproteomic Analysis

Six samples were analyzed, including 3 HUVECs that were exposed to CLL-EVs (herein HUVEC-exposed) and 3 unexposed HUVECs (herein HUVEC-unexposed). For each sample, a total of 1.4 × 10^6^ cells were washed twice with ice-cold PBS and lysed with 8 M urea, 40 mM Tris/HCl (pH 7.6), 1× EDTA-free protease inhibitor mixture (Complete Mini, Roche), and 1× phosphatase inhibitor mixture (Sigma). The lysate was centrifuged at 20,000 rpm for 1 h at 4 °C. 

The protein concentration was determined using the Bradford method (Coomassie (Bradford) Protein Assay Kit, Thermo Scientific). The supernatant was reduced with 10 mM DTT at 56 °C for 1 h and alkylated with 25 mM iodoacetamide for 45 min at room temperature in the dark. The protein mixture was diluted with 40 mM Tris/HCl to a final urea concentration of 1.6 M. We induced digestion by adding sequencing-grade trypsin (Promega, Mannheim, Germany; 1:100 enzyme: substrate ratio) and allowing samples to incubate at 37 °C for 4 h. Subsequently, another 1:100 quantity of trypsin was added for overnight digestion at 37 °C. Samples were acidified with TFA to pH 2 to stop the trypsin activity. SepPack columns (C18 cartridges, Sep-Pak Vac, 1 cc (50 mg), Waters Corp., Eschborn, Germany; solvent A, 0.07% TFA; solvent B, 0.07% TFA, 50% ACN) were used for peptide desalting according to the manufacturer’s instructions, and eluates were dried and stored at −80 °C. 

To enrich for phosphopeptides, the samples were loaded into embedded media tips (NuTips, glysci) supplemented with TiO_2_, according to the manufacturer’s recommendation. Eluted phosphopeptides were dried by vacuum centrifuge and stored at −20 °C until further analysis. 

ULC/MS-grade solvents were used for all chromatographic steps. Each sample was loaded using split-less nano-ultra-performance liquid chromatography (10 kpsi nanoAcquity; Waters, Milford, MA, USA). The mobile phase was as follows: (A) H_2_O + 0.1% formic acid and (B) acetonitrile + 0.1% formic acid. Desalting of the samples was performed online using a reversed-phase Symmetry C18 trapping column (180 µm internal diameter, 20 mm length, 5 µm particle size; Waters). The peptides were then separated using a T3 HSS nano-column (75 µm internal diameter, 250 mm length, 1.8 µm particle size; Waters) at 0.35 µL/min. Peptides were eluted from the column into the mass spectrometer using the following gradient: 4% to 25%B in 155 min, 25% to 90%B in 5 min, maintained at 90% for 5 min, and then back to initial conditions.

The nanoUPLC was coupled online through a nanoESI emitter (10 μm tip; Fossil, Spain) to a quadrupole orbitrap mass spectrometer (Q Exactive Plus Thermo Scientific) using a FlexIon nanospray apparatus (Thermo Scientific). 

Data were acquired in data-dependent acquisition mode, where the MS1 mass range was 300–1650 *m*/*z*, the resolution was 70,000, the automatic gain control (AGC) was set to 3 × 106 and the injection time was 50 ms. The MS2 resolution was set to 17,500, the AGC target was 1e5, and the maximum injection time was set to 120 ms. 

Raw data were imported into Expressionist^®^ software version 10.5 (Genedata, Switzerland) and processed as described here. The software was used for retention time alignment and peak detection of precursor peptides. A master peak list was generated from all MS/MS events and sent for a database search using Mascot v2.5.1 (Matrix Sciences, Columbus, OH, USA). Data were searched against the human sequences in UniprotKB (http://www.uniprot.org/, accessed on 15 October 2018) appended with common laboratory contaminant proteins. Fixed modification was set to carbamidomethylation of cysteines, and variable modifications were set to oxidation of methionines, deamidation of N or Q, and phosphorylation of S, T, or Y. The search results were then filtered using the PeptideProphet algorithm to achieve a maximum false discovery rate of 1% at the protein level. Peptide identifications were imported back into Expressionist to annotate identified peaks. The raw data are available through the MassIVE repository with accession MSV000092918 (https://massive.ucsd.edu/ProteoSAFe/static/massive.jsp, accessed on 15 October 2018) [9]. User: MSV000092918_reviewer Password: uziel1324).

The intensity distributions of each sample are shown in the Appendix A. 

Annotation analysis was carried out in 2 steps. We used DAVID (https://david.ncifcrf.gov/ accessed on 15 October 2018) [12] to prioritize the upregulated phospho-proteins in HUVEC-exposed cells according to biological significance. With this tool, cluster analysis identified the most enriched processes within the Gene Ontology (GO) database. The group of proteins that were included in the most enriched process was used as the “anchor protein group” for pathway analysis using the Advanced Network Analysis Tool (ANAT). The network that is formed by the algorithm of ANAT is built around a specific set of nodes termed “anchor set” [13]. 

### 2.13. Cell Transfection

To generate endothelial cells that overexpress β-catenin, we used a CTNNB1-expressing plasmid, which encodes for the β-catenin gene (Addgene, Watertown, MA, USA). We used the jetPEI^®^ transfection reagent (polypus transfection, Strasbourg (France)) to transfect both HUVECs and HEAOCs with this construct according to the manufacturer’s instructions. Briefly, a total of 10^5^ cells were plated with 6 μg of DNA mixed with 12 μL of DNA transfection reagent for 72 h. To confirm efficient transfection, we transfected in parallel under similar conditions with GFP-expressing plasmids (Addgene). 

### 2.14. Chromatin Immunoprecipitation (ChIP) Assay

The binding of several transcription factors, including nuclear factor kappa-light-chain-enhancer of activated B cells (NFκB), lymphoid enhancer-binding factor (LEF), and CCAAT/enhancer-binding protein (CEBP), to the promoter of IL-6 in HUVEC cells was assessed by chromatin immunoprecipitation (ChIP) assay according to the manufacturer’s instructions (EZ-ChIP; Millipore, Bedford, MA, USA). Briefly, we added formaldehyde to the culture to a final concentration of 1% at room temperature for 10 min to cross-link the DNA binding proteins to the chromatin, and the reaction was stopped by adding glycine to a final concentration of 125 mM. Cells were washed in PBS and resuspended in 1 mL lysis buffer containing protease inhibitor and RNAse cocktail. Cross-linked DNA was then sheared by sonication to produce a fragment of ≈200–1000 base pairs in length. Following sonication, the samples were centrifuged, and the supernatant was diluted 10 times in a dilution buffer containing a protease inhibitor cocktail. Protein G Agarose beads were used to pre-clear the diluted chromatin for 1 h at 4 °C on a rotating platform. From the pre-cleared chromatin, 1% was saved as input, and the rest was incubated with anti-NFkB, anti-LEF, and anti-CEBP antibodies (all purchased from Abcam). After overnight incubation at 4 °C on a rotating platform, the antibody–protein–DNA complexes were collected by adding Protein G Agarose beads and incubating for 1 h at 4 °C with rotation. The DNA was then reverse-cross-linked at 65 °C. RNA was degraded by RNAse (QIAquick PCR purification, Qiagen, Hilden, Germany) and proteins by proteinase K (Thermo Fisher Scientific, Waltham, MA, USA). We then performed quantitative PCR on purified DNA using the following primers for NFkB p65:
Forward: 5′-CTACTTCCCTCCCAAGATGCC-3′Reverse: 5′-TGCCCTGCTGTGTTTCTCATT-3′LEF/TCF distal:Forward: 5′-TCCAACATCAGCTGGCTCTTT-3′Reverse: 5′-TTGAGTCCCACTGAGCAGACA-3′LEF/TCF proximal:Forward: 5′-GAGCGATAAACACAAACTCTGCAA-3′Reverse: 5′-CTGGAGCCCTGAAATTACTGAAG-3′CEBP: Forward: 5′-GCAAAGAAACCGATTGTGAAGG-3′Reverse: 5′-CCTTGCACAACACCAAAACACT-3′

### 2.15. Enzyme-Linked Immunosorbent Assay (ELISA) for the Detection of IL-6

HUVEC secretion of Interleukin 6 (IL-6) was measured by a Human IL-6 Quantikine ELISA Kit (R&D, Minneapolis, MN, USA) according to the provided protocol. Briefly, we removed HUVECs from the cultured medium by centrifugation of our samples at 1000 rpm for 5 min. Then, we incubated the supernatant in the presence of an anti-IL6 antibody for 2 h and used a washing buffer to remove the unbound proteins. After 4 washes, we added a different anti-IL6 conjugated antibody for 2 h. Then, we washed the unbound antibody and added the substrate for enzymatic cleavage in each well for 20 min incubation in dark conditions. Subsequently, we added a stop solution in each well for 5 min and used a spectrophotometer to measure the optical density of our sample and of a serial of standard calibration diluents. The microplate reader (Epoch, Biotek, Winooski, VT, USA) was set at 450 nm. 

### 2.16. Annexin V Assay

Apoptosis was evaluated by Annexin V and PI staining (MEBCYTO^®^ Apoptosis Kit MBL, Nagoya, Japan) according to the manufacturer’s instructions. Briefly, cells were labeled by Annexin V and propidium iodide. After the incubation step, the fluorescent cells were analyzed by flow cytometry and the data were analyzed using Kaluza Acquisition Software 3.1. Apoptosis was evaluated by Annexin V and PI staining (MEBCYTO^®^ Apoptosis Kit MBL, Nagoya, Japan) according to the manufacturer’s instructions. Briefly, cells were labeled by Annexin V and propidium iodide. After the incubation step, the fluorescent cells were analyzed by flow cytometry and the data were analyzed using Kaluza Acquisition Software. 

### 2.17. Statistical Analysis

Data are presented as median + range for continuous data and frequency (percentage) for categorical variables. To analyze levels of IL-6 in a cultured medium, we used a two-way, repeated measures analysis of variables (ANOVA) with EV exposure as “between” and time as “within” variables. The analysis was performed using the GraphPad Prism 8 version.

## 3. Results

### 3.1. CLL-EVs Are Taken up by Endothelial Cells

To isolate CLL-EVs, we harvested PBMCs from 45 treatment-naïve patients. At least 88% of the cells were CD19/CD5-positive, indicating that they are CLL cells. We grew these cells in EV-depleted media for 72 h and collected the EVs from the conditioned medium using differential centrifugation. By nanoparticle tracking analysis (NTA), we identified the presence of a large amount of 80–140 nm particles (Figure 1A), and by EM, we found vesicles the size of EVs (Figure 1B). Western blotting analysis confirmed that our EVs possess the known EV markers (CD81, CD9, CD63, as well as TSG101) but lack the non-EV marker, Calnexin (Figure 1C). Flow cytometry confirmed that more than 50% of these particles are CD81-positive, indicating that they are EVs, and more than 50% of these particles are CD19-positive, indicating that they are derived from CLL cells (Figure 1D). Together, these experiments verified that we can isolate CLL-derived EVs from patients. Then, we exposed HUVECs to an increasing amount of CLL-EVs that were double-stained with the membrane red dye FM-1-43 and FITC-conjugated peptide that penetrates the EVs’ membrane. By confocal microscopy, we detected the stained EVs inside the cells (Figure 1D), and by flow cytometry, we confirmed the confocal data and showed that CLL-EVs enter HUVECs in a dose-dependent manner (Figure 1E). Kinetics studies showed that the maximal uptake occurs after 24 h (Figure 1F). Notably, higher levels of IL-6 were detected in patients with CLL compared to their healthy individual counterparts [15]. Assuming that the difference between CLL patients and healthy people is the characteristics of B cells—neoplastic ones in the case of CLL patients versus normal B cells in healthy populations—we have used this information as indirect evidence for the possible control of the specificity of CLL-derived EVs’ effect on the secretion of IL-6 by endothelial cells. 

### 3.2. CLL-EVs Induce Phosphorylation of β-Catenin in Hosting Endothelial Cells

Previous studies have shown that neoplastic EVs are capable of modulating the cellular activity of hosting cells by inducing protein phosphorylation [13,16,17]. Therefore, we sought to explore the role of CLL-EVs in modulating the protein phosphorylation landscape of hosting endothelial cells. To do this, we adopted a high-throughput approach and used mass spectrometry to identify phosphorylated proteins prior to and after exposing HUVECS to CLL-EVs. With this approach, we identified 4522 phosphopeptides, of which 53 phospho-proteins were at least 2-fold upregulated, and none were downregulated in HUVEC-exposed cells (Table 1). Pathway analysis unraveled the central position of β-catenin in a network that included all 53 upregulated phospho-proteins (Figure 2A). Our proteomics data did not reveal proteins of the WNT family ligands of the β-catenin pathway, including Wingless/Integrated (WNT)5A, WNT3A, and WNT1. Of the β-catenin regulated gene tested, we detected the proto-oncogene transcription factor cmyc and the TFC4. These proteins were detected both in EV-treated and in control non-treated HUVECs with no statistically significant difference. Other β-catenin target genes, including CCND1, PPARD, ASCL2, ABCB1, and BIRC, were not detected.

To validate the proteomics results, we selected three phospho-proteins: β-catenin, CD11B, and WBP11, for which the phospho isoform was 3.6-fold (for β-catenin) to over 4700-fold (for CD11) higher in EV-exposed endothelial cells. We chose β-catenin based on its central role according to the pathway analysis, WBP1 as a representative of a nuclear protein, and CD11 as a representative of a membranal protein. 

After immunoprecipitation of the phosphorylation isoforms, we detected by Western immunoblotting an increased phosphorylated to total protein ratio in all three proteins tested (Figure 2B). To determine whether the increase in levels of p-β-catenin comes from newly formed β-catenin or increased phosphorylation of pre-existing molecules, we performed Western immunoblotting using EVs from three additional patients. As shown in Figure 2C, levels of β-catenin remained similar in HUVEC-exposed and in HUVEC-non-exposed cells, indicating that CLL-EVs induced phosphorylation rather than the generation of newly formed β-catenin molecules. 

### 3.3. Upregulation of β-Catenin Induces IL-6 Secretion from Hosting Endothelial Cells

Previous studies showed that neoplastic cells use EVs to recruit bystander cells, which in this way become tumor-supportive cells [13,14,18,19]. Whether and how CLL cells use this strategy to recruit endothelial cells is unknown. Since we found that CLL-EVs are taken up by endothelial cells and induce upregulation of β-catenin, we hypothesized that the activated β-catenin axis in endothelial cells is beneficial to their parental neoplastic cells.

Upstream of the transcription starting site of the IL-6 gene, there are four binding sites to three transcription factors that are activated by β-catenin and span the 5Kbp IL-6 promoter site. This includes two binding sites to LEF/TCF and CEBP and NFκB binding sites (Figure 3A top panel). Since in CLL cells, IL-6 promotes CLL cells’ survival, we postulated that by activating the β-catenin pathway, endothelial cells become “micro-factories” for the production of humoral IL-6. In this way, endothelial cells become “CLL-supportive cells”. To verify the role of β-catenin in promoting the production of IL-6 molecules in endothelial cells, we transfected HUVECs with β-catenin containing plasmid, and after 48 h, we verified by Western Immunoblotting that the protein levels of β-catenin were upregulated (Figure 3B). Within this short timeframe, there were no differences in viability or proliferation rates between transfected and non-transfected cells. Then, by ChIP assay, we found that in HUVECs that were transfected with this plasmid, the binding of three different transcription factors to the IL-6 promoter was markedly increased. Specifically, the binding of LEF/TCF to its proximal binding site was 28.4-fold increased, and to its distal binding site, it was 12.3-fold increased. Likewise, the binding of CEBP was 3.1-fold increased, and NFκB binding was 1.5-fold increased. (Figure 3C). Taken together, our data show that through the β-catenin axis, the IL-6 promoter is activated in HUVECs. To show that activation of this pathway leads to IL-6 secretion, we used ELISA. With this assay we found that levels of IL-6 were markedly increased in the medium of HUVECs that were exposed to CLL-EVs (Figure 3D).

### 3.4. Endothelial-Cell-Induced IL-6 Drives STAT3 Phosphorylation and Leads to Decreased Rate of Apoptosis of CLL Cells 

We have previously shown that IL-6 induces tyrosine phosphorylation of CLL cells [20]. As shown in Figure 4A, we confirmed this observation on three patients with CLL. Next, we grew endothelial cells in the presence of CLL-EVs that are now enriched with IL-6 (Figure 3D) and subsequently grew CLL cells in the same medium. As shown in Figure 4B (upper panel), we observed a dose-dependent increase in tyrosine phosphorylation of STAT3 and, as shown in Figure 4B (lower panel), a dose-dependent decrease in apoptosis.

## 4. Discussion

Here, we show that CLL cells shape their own fate. They do so by releasing EVs that turn endothelial cells into IL-6-producing cells. IL-6, in turn, provides the neoplastic cells with a survival advantage (Figure 5). 

CLL cells reside in lymph nodes, bone marrow, and peripheral blood. In these tissues, CLL cells come into proximity with mesenchymal stromal cells and various immune cells. Since blood vessels are scattered within these tissues, endothelial cells are also ubiquitously present. In line with previous studies [12,13], we show here that EVs are taken up by endothelial cells in a dose- and time-dependent manner. 

That EVs alter the cellular makeup of hosting cells has been shown in various cell types and across different molecular settings. For example, EVs from virally infected cells of injured liver can turn hematopoietic progenitors into hepatocytes [21]. These alterations are specific to the type of donor cells [22,23]. Hence, we show here alterations in the phosphorylation status of endothelial cells after exposing the cells to CLL-derived EVs. In another set of experiments, we received a completely different set of proteins that were phosphorylated after exposing the same endothelial cells to monocyte-derived EVs.

Which part of the EV cargo leads to the phosphorylation of β-catenin is unknown to us. In fact, we encountered only a few studies that addressed similar questions. There is evidence that functional proteins are transported via cancer-derived EVs. For example, Song et al. showed that cancer-cell-derived EVs transport functional tyrosine kinase receptors [16]. Likewise, anti-GFP loaded on CLL-EVs successfully silenced GFP expression in recipient cells [20].

In cancer, neoplastic-derived EVs mediate tumor–tumor and tumor–environment communication [21]. For example, we have shown that neoplastic EVs transform bystander fibroblasts into cancer-associated fibroblasts (CAFs) [24]. By initiating the remodeling of the extracellular matrix and secreting pro-angiogenic chemokines, CAFs promote and maintain tumor growth over time [25]. Like CAFs, tumor-associated endothelial cells (TAECs) that control the passage of nutrients into the surrounding of solid tumors differ significantly from normal blood vessels in morphology, gene expression, and functionality [26]. By trafficking between tumor and endothelial cells and unloading a cargo of pro-angiogenic factors, EVs secreted by tumor cells actively induce the formation of TAECs in solid tumors [27,28,29]. Accelerated angiogenesis was also described across a wide range of hematological malignancies, including CML [30], AML [31], MDS [32], multiple myeloma [33], and CLL [20]. Whereas in solid tumors the formation of blood vessels is required to feed the growing mass, what is the benefit from newly formed blood vessels in liquid tumors composed of discrete cells is not obvious. Here, we show that it may not be the blood vessels that are important but the endothelial cells that, under the influence of CLL cells, become IL-6-producing cells. 

Discovered in 1986 [29], IL-6, previously B-cell-inducing factor 228, is a pleiotropic cytokine with numerous activities in diverse biological processes, such as embryonic development [31], bone metabolism [32], and hematopoiesis [33]. IL-6 is mostly formed in non-immune cells, but its presence promotes the recruitment and activation of immune response [16]. Here, we show that this is not a unidirectional process. Rather, there is a crosstalk between the neoplastic and non-neoplastic cells. In our study, CLL cells signal endothelial cells through EV release to produce and secrete IL-6. Proteomics analysis indicated that the phosphorylated fraction of β-catenin is significantly upregulated in endothelial cells that were exposed to CLL-exomes; previous studies showed that in these cells, β-catenin binds and activates NFκB [20]. In another study, Rickman et al. showed that disturbed flow increases endothelial inflammation and permeability via a Frizzled-4-β-catenin-dependent pathway [34]. This is in line with our ChIP results showing that β-catenin enhances the activity of NFκB in HUVECs. In fact, our data showed that even a moderate increase in β-catenin had a consistent and reproducible effect on the binding of β-catenin and other transcription factors to the IL-6 premotor site. Intriguingly, the phosphorylated and unphosphorylated β-catenin isoforms activate distinct genetic programs. While the unphosphorylated isoforms primarily activate the Wnt pathway [35], in our study, the phosphorylated form activated an inflammatory response (e.g., IL-6). 

One drawback of our study concerns the issue of specificity of the CLL-cell-derived EVs. Our control throughout the whole study was endothelial cells that were not exposed to any type of EVs. However, we have not exposed these endothelial cells to normal B-cell-derived EVs. Luckily, we have an “in vivo” indirect answer for this point. Hulkkonen et al. from the CLL study group have recently shown that the levels of IL-6 in patients with CLL are much higher than in healthy individuals [15]. This may serve as indirect proof that CLL-EVs initiate the whole cycle of IL-6 induction in endothelial cells, which eventually protects the neoplastic cells from apoptosis, as we have shown here.

In summary, CLL cells are long-lived lymphocytes. Longevity is maintained by various mechanisms that the cells adopt. In an ever-expanding clone, these mechanisms may also emerge with time and, therefore, are not shared by all clonal cells. In some, the constitutive activation of pro-survival pathways may promote longevity without the need for external cues [13]. In others, these cues are likely needed. Here, we describe how, by EV-mediated cell-to-cell communication, CLL cells may actively shape the environment in a way that ensures their long-lived existence. 

## Figures and Tables

**Figure 1 biomedicines-12-01381-f001:**
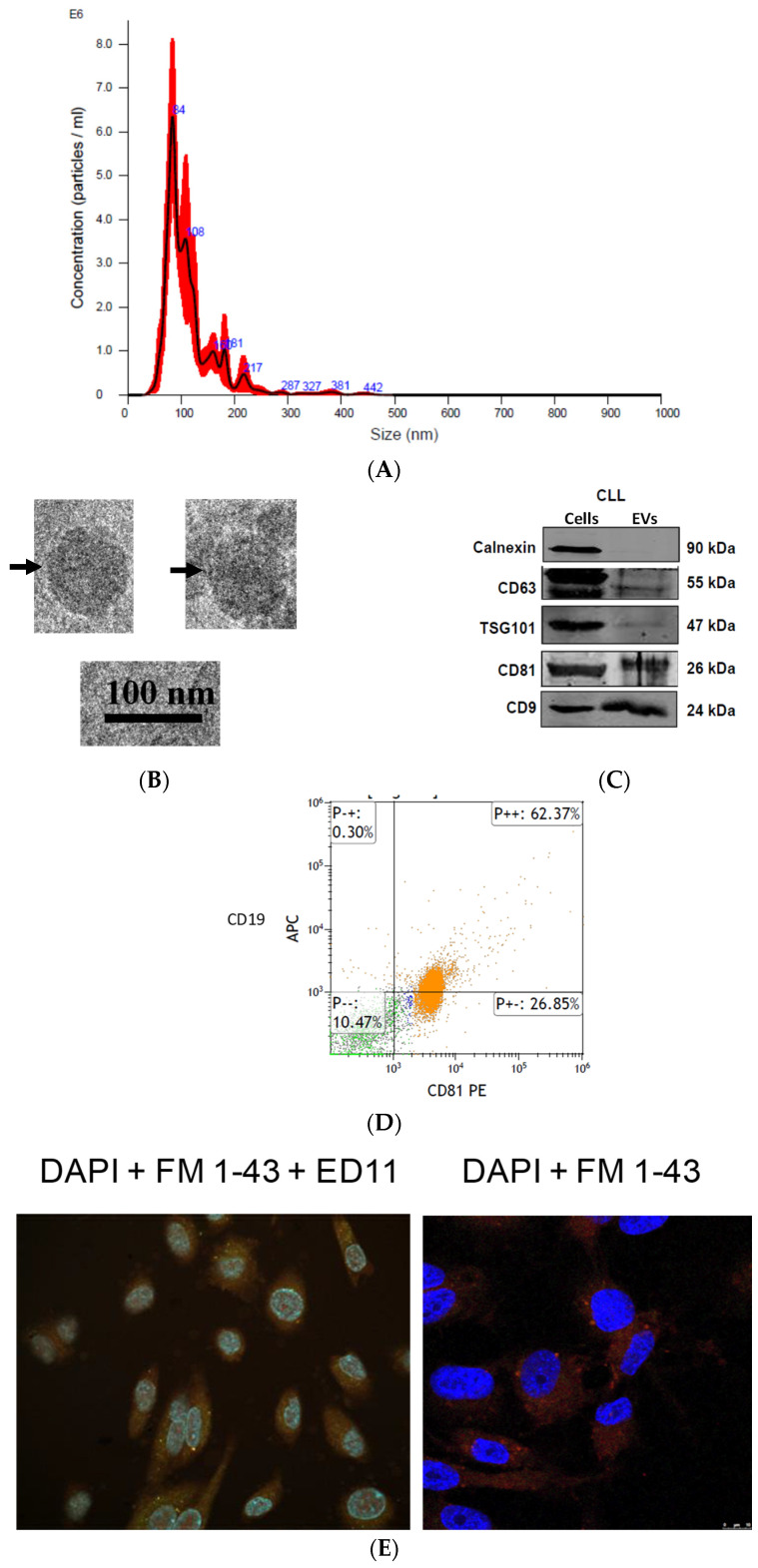
CLL-EVs are taken up by endothelial cells. (**A**) CLL cells were grown in an *EV*-depleted medium for 72 h, and *EVs* were isolated by ultracentrifugation. Shown is a typical representation of a nanoparticle tracking analysis with highest concentration of particles ranging in size between 80 and 140 nm. (**B**) Transition electron microscopy image of CLL-derived EVs. Arrows depict the typical cap-shaped *EVs*. (**C**) Western blot of protein extracts from CLL cells (right) and CLL-derived EVs left). (**D**) Flow cytometry of *EVs* derived from CLL cells. As shown, 62% of the nanoparticles were double-stained for CD19 (CLL marker) and CD81 (EV marker). (**E**) HUVECs were incubated for 24 h in the presence of CLL-*EVs* that were double-stained with the red dye FM-143 and with the green FITC-conjugated ED11 peptide. DAPI staining (blue) of the cell nuclei is shown in each panel. Fluorescent image (left) and confocal image (×63) (right), both showing *EVs* within HUVECs. (**F**) HUVECs (CD5 negative cells) were exposed to an increasing dose of CLL-*EVs* (CD5 positive *EVs*). Shown here is a dose-dependent increase in the rate of CD5 + HUVECs. (**G**) The kinetics of CLL-EV uptake by HUVECs. HUVECs were exposed to 2 × 10^9^ of CLL-*EVs* stained for the red membranal FM1-43 dye. The rate of FM1-43-positive cells was determined by flow cytometry at 5 time points. Shown is a time-dependent increase in FM1-43-positive cells. Maximal uptake occurred after 24 h. Experiments (**C**–**E**) were conducted in triplicates. (**H**) Dose- (**left**) and time-dependent (**right**) uptake of *EVs* by HUVECs.

**Figure 2 biomedicines-12-01381-f002:**
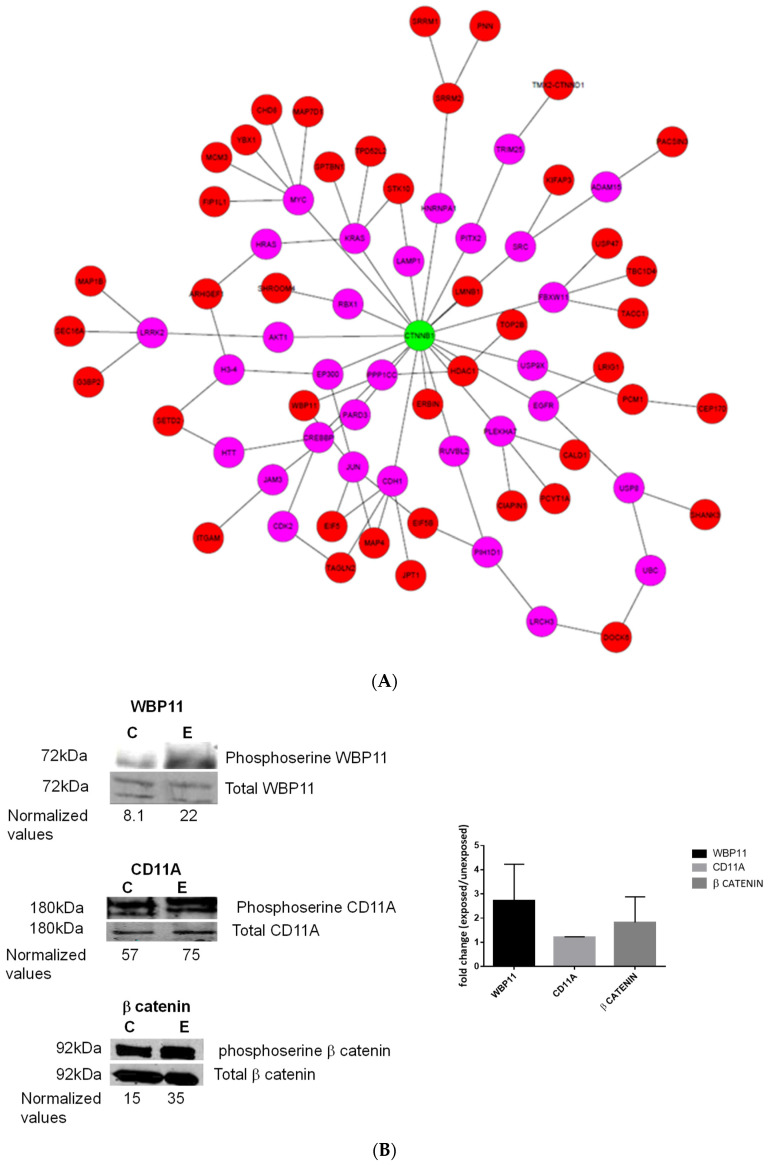
CLL-EVs induce phosphorylation of β-catenin in hosting endothelial cells. (**A**) A protein network generated using the ANAT algorithm based on the 53 phospho-proteins that were upregulated in HUVECs exposed to CLL-EVs (Table 1). The algorithm finds a compact network connecting the anchor (β-catenin outlined in green) to the hit proteins (outlined in red) based on a protein–protein interaction network database. Shown is the central role of β-catenin, connected to a set of upregulated phospho-proteins through a series of linking proteins (pink). (**B**) Validation of the proteomics results for 3 of the 53 upregulated phospho-proteins. The cell lysate of HUVECs that were exposed to CLL-EVs was immunoprecipitated with WBP11 (**upper panel**), CD11A (**middle panel**), and β-catenin (**lower panel**) antibodies. Western blot analysis of the phospho isoform of each protein is shown prior to (marked as C) and after (marked as E) exposure to CLL-EVs. Standardized densitometry values of three independent experiments are presented in the right panel. As shown, the fraction of the phosphorylated isoform was 1.25- to 2.75-fold increase in HUVEC-exposed cells. (**C**) Western Immunoblot of total β-catenin isolated from the cells of three different endothelial cells isolates. As shown, there was no difference in the total levels of β-catenin in response to CLL-derived EV exposure in these cells. All experiments were conducted in triplicates.

**Figure 3 biomedicines-12-01381-f003:**
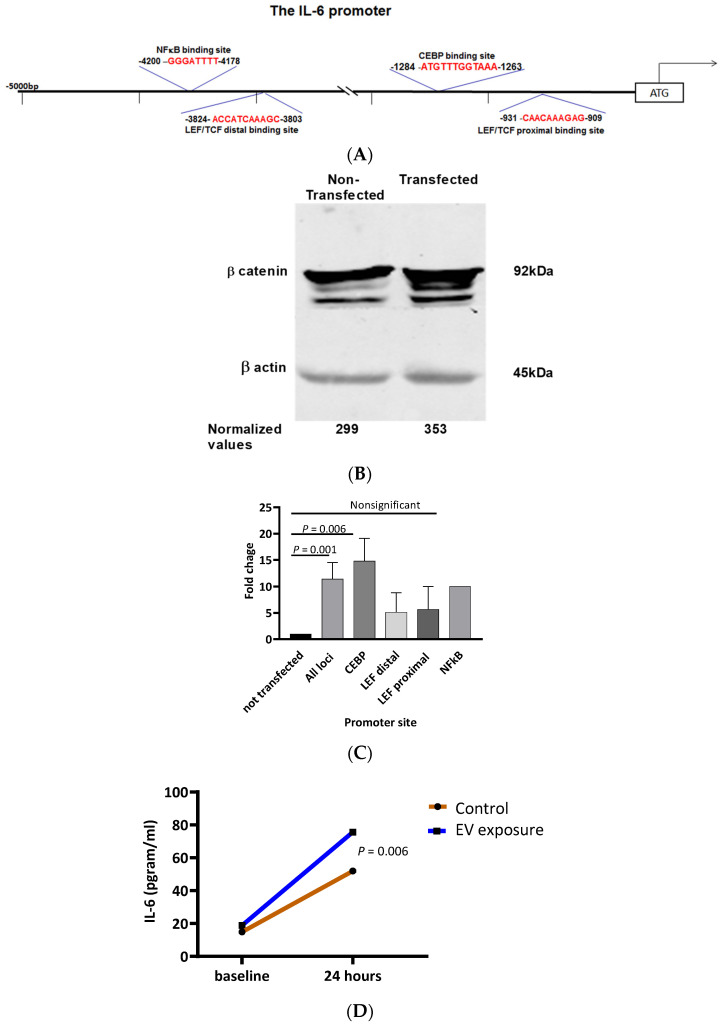
Upregulation of β-catenin induces IL-6 secretion from hosting endothelial cells. (**A**) Schematic diagram of the 5KB DNA sequence upstream of the IL-6 5′ transcription start site. (**B**) HUVECs were transfected with a β-catenin-containing plasmid, and its level was assessed by Western immunoblotting 5 min after exposure. Below each lane are densitometry values of β-catenin relative to levels of loading control. (**C**) Quantification of the levels of transcription factors binding to the IL-6 promoter. The binding site for 2 LEF/TFC, CEBP, and NFκB transcription factors are highlighted in red. (**D**) Levels of IL-6 secretion by endothelial cells in response to CLL-derived *EVs*. All experiments (**B**–**D**) were conducted in triplicates.

**Figure 4 biomedicines-12-01381-f004:**
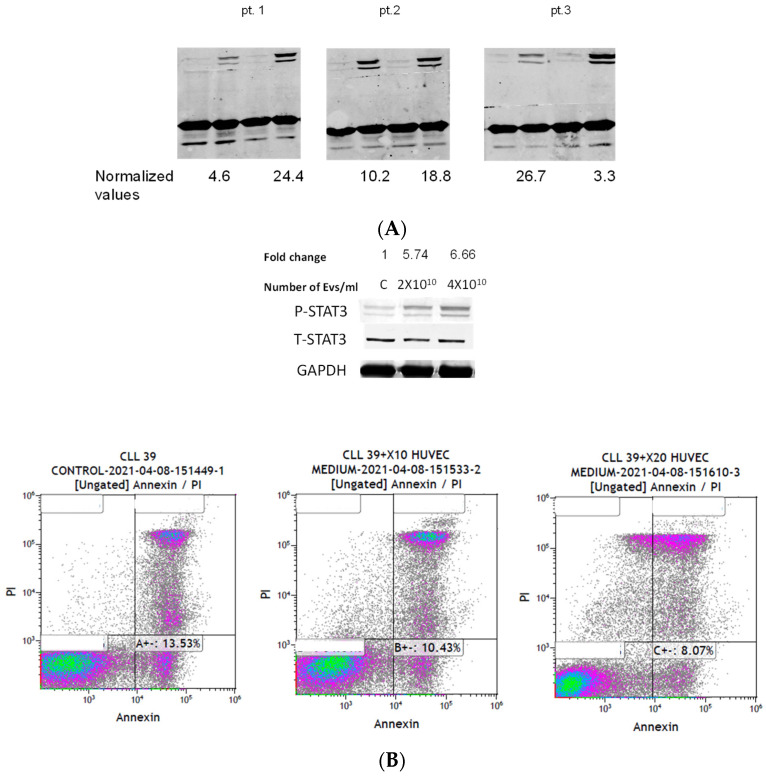
Endothelial-induced IL-6 drives STAT3 phosphorylation and leads to decreased apoptosis rates of CLL cells. (**A**) Freshly isolated CLL cells from 3 patients were incubated with 20 µg of IL-6. After 30 min the cells were harvested, the protein extracted, and subjected to Western immunoblotting. As shown, IL-6 induced phosphorylation of STAT3 in CLL cells. (**B**) Endothelial cells were exposed to 2 × 10^10^ or 4 × 10^10^ CLL-derived EVs or left unexposed. After 72 h, the endothelial cells were filtered out. Shown are CLL cells that were grown in this medium. This experiment was repeated twice. Similar to CLL cells that were directly exposed to IL-6, there was a dose-dependent phosphorylation of STAT3 (**upper panel**). As shown by Annexin/PI staining, there was a dose-dependent decrease in early apoptosis (**lower panel**). This experiment was repeated three times.

**Figure 5 biomedicines-12-01381-f005:**
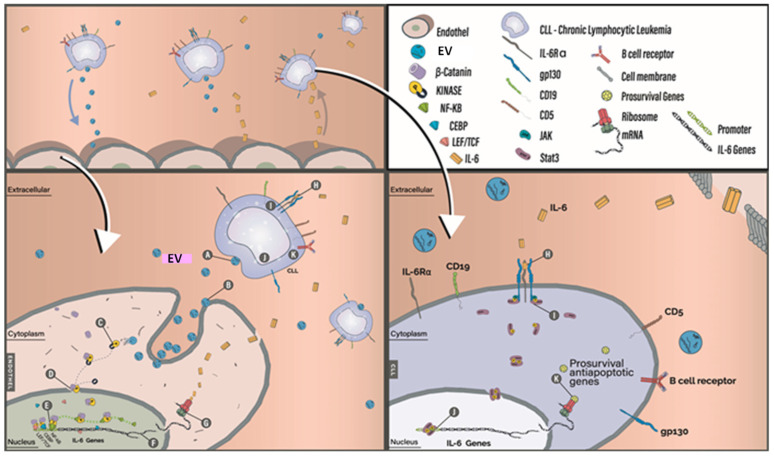
A model of the crosstalk between endothelial cells and CLL cells. Upper left panel: A cross-section of a blood vessel. CLL cells secrete EVs. Endothelial cells take up CLL-derived EVs and, as a result, produce and secrete IL-6. Subsequently, IL-6 binds the IL-6 receptor on the cell membrane of CLL cells. Bottom left panel: a cross-section of an endothelial cell. (A) CLL cells release EVs; (B) EVs are taken up by endothelial cells through membrane fusion; (C) levels of phosphor-β-catenin are upregulated in exposed cells; (D) phosphor-β-catenin is translocated to the nucleus; (E) transcription factors preferentially bind phosphor-β-catenin which increases their affinity to DNA. As a result, NFκB p65, LEF/TCF, and CEBP bind the IL-6 promoter; (F) NFκB p65, LEF/TCF, and CEBP binding to the IL-6 promoter induces transcription of IL-6 transcripts; (G) translation of IL-6 transcripts and secretion of IL-6 protein. Bottom right panel: A cross-section of a CLL cell. (H) IL-6 binds the IL-6 receptor, and two adjacent receptors dimerize; (I) STAT3 is phosphorylated on tyrosine 705 residue and dimerizes and translocates to the nucleus; (J) STAT3 induces transcription of pro-survival and anti-apoptotic genes.

**Table 1 biomedicines-12-01381-t001:** Baseline patient characteristics (N = 45).

Characteristics	Data
**median age** Years (range)	71 (52–89)
**median white blood cell counts** ×10^9^/L (range)	66 (14–292)
**median absolute lymphocyte count** ×10^9^/L range)	54 (9–281)
**Rai stage** N (%)	
Rai = 0	27 (60)
Rai = 1	7 (16)
Rai = 2	6 (13)
Rai = 3	1 (2)
Rai = 4	4 (9)
**β2M** (mg/L) N (%)	
<4	10 (30)
>4	23 (70)
Not taken	12
**IGHV mutation** N (%)	
Mutated	10 (22)
Unmutated	20 (44)
unknown	15 (33)
**FISH result** N	
Del17p/11q/tri12/13q/negative/unknown	3/7/6/15/4/19
**Comorbidities** N (%)	
ischemic heart disease	10 (22)
Diabetes Mellitus	9 (20)
hypertension	4 (9)
cerebrovascular accident	0 (0)
**Patients with other malignancies** N (%)	9 (20)
**Type of other malignancies *** (N)	6
**Mean ejection fraction** % (S.D.)	58 (9.7)
**Drugs** N (%)	
Aspirin	18 (40)
Statins	20 (44)
Metformin	7 (15)

* b2M, beta-2 microglobulin; FISH, fluorescence in situ hybridization; Del, deletion; Tri, trisomy. Carcinoma of the breast (N = 6), melanoma (N = 2), carcinoma of the lung (N = 2), acute myelocytic leukemia (N = 1), carcinoma of the prostate (N = 1), carcinoma of the stomach (N = 1).

## Data Availability

No new data were created in this study.

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
