# Peer review of "Chronic Lymphocytic Leukemia (CLL)-Derived Extracellular Vesicles Educate Endothelial Cells to Become IL-6-Producing, CLL-Supportive Cells"

_biomedicines, 2024, doi:10.3390/biomedicines12071381_

Round 1

Reviewer 1 Report (Previous Reviewer 2)

Comments and Suggestions for Authors

I have carefully reviewed the manuscript titled "Chronic lymphocytic leukemia (CLL) derived Extracellular Vesicles educate endothelial cells to become IL-6 producing, CLL supportive cells." The study explores the hypothesis that extracellular vesicles (EVs) released by CLL cells can induce a transformation of endothelial cells into supportive cells for CLL. The authors aimed to investigate this hypothesis by treating vein-derived (HUVECs) and arterial-derived (HAOEC) endothelial cells with EVs isolated from treatment-naïve CLL patients.

Overall, the study provides evidence that CLL cells communicate with endothelial cells through EVs, leading to the transformation of endothelial cells into IL-6 producing cells. This interaction mediated by EVs may contribute to the creation of a supportive microenvironment for CLL progression. However, I would like to address a few points for consideration:

  1. Terminology: The manuscript uses the term "exosomes" to describe the EVs. It is important to note that cells secrete various types of EVs, including exosomes, microvesicles, small EVs, oncosomes, and apoptotic bodies. To ensure accuracy, it is recommended to avoid specific terminology such as "exosomes" unless the mechanisms of biogenesis and secretion have been thoroughly investigated and confirmed in this study.

  2. Figure 1C: There seems to be a mistake in the description of the gel image, as well as the labeling of the cells on the left and the exosome on the right. It is crucial to rectify this error to ensure clarity and accurate representation of the experimental results.

  3. Figure 2A: The protein network image appears to be of poor quality. I suggest that the authors enhance the image quality to improve its clarity and ensure the accurate visualization of the protein network.

  4. Line 594: The phrase "Please add" in line 594 should be removed to ensure the clarity and conciseness of the manuscript.

Overall, the manuscript provides valuable insights into the communication between CLL cells and endothelial cells mediated by EVs. Addressing the mentioned points will further improve the accuracy and presentation of the research findings. I recommend the authors to carefully revise and finalize the manuscript accordingly.

Author Response

1st reviewer

I have carefully reviewed the manuscript titled "Chronic lymphocytic leukemia (CLL) derived Extracellular Vesicles educate endothelial cells to become IL-6 producing, CLL supportive cells." The study explores the hypothesis that extracellular vesicles (EVs) released by CLL cells can induce a transformation of endothelial cells into supportive cells for CLL. The authors aimed to investigate this hypothesis by treating vein-derived (HUVECs) and arterial-derived (HAOEC) endothelial cells with EVs isolated from treatment-naïve CLL patients.

Overall, the study provides evidence that CLL cells communicate with endothelial cells through EVs, leading to the transformation of endothelial cells into IL-6 producing cells. This interaction mediated by EVs may contribute to the creation of a supportive microenvironment for CLL progression. However, I would like to address a few points for consideration:

  1. Terminology: The manuscript uses the term "exosomes" to describe the EVs. It is important to note that cells secrete various types of EVs, including exosomes, microvesicles, small EVs, oncosomes, and apoptotic bodies. To ensure accuracy, it is recommended to avoid specific terminology such as "exosomes" unless the mechanisms of biogenesis and secretion have been thoroughly investigated and confirmed in this study.

OK, we have replaced "exosomes" with "EVs" throughout the manuscript.

 Figure 1C: There seems to be a mistake in the description of the gel image, as well as the labeling of the cells on the left and the exosome on the right. It is crucial to rectify this error to ensure clarity and accurate representation of the experimental results.

We thank the reviewer for this note, but actually there is no mistake here… on the left there are exosomal markers present in the neoplastic CLL cells while on the right we can see these markers in their cognate secreted exosomes. Indeed, calnexin, which is not present in exosomes due to its presence in the ER, is missing here at the exosomal extracts lane. See for example: Masaki K, Ahmed ABF, Ishida T, Mikami Y, Funabashi H, Hirota R, Ikeda T, Kuroda A. Chromatographic purification of small extracellular vesicles using an affinity column for phospholipid membranes. Biotechnol Lett. 2023 Dec;45(11-12):1457-1466. Or: Mathieu, M., Névo, N., Jouve, M. et al. Specificities of exosome versus small ectosome secretion revealed by live intracellular tracking of CD63 and CD9. Nat Commun 12, 4389 (2021).

 Figure 2A: The protein network image appears to be of poor quality. I suggest that the authors enhance the image quality to improve its clarity and ensure the accurate visualization of the protein network.

We have replaced the figure with a more accurate and clear one.

 Line 594: The phrase "Please add" in line 594 should be removed to ensure the clarity and conciseness of the manuscript.

Correct! This has been removed.

Overall, the manuscript provides valuable insights into the communication between CLL cells and endothelial cells mediated by EVs. Addressing the mentioned points will further improve the accuracy and presentation of the research findings. I recommend the authors to carefully revise and finalize the manuscript accordingly.

We sincerely thank the reviewer for a careful reading of the manuscript and raising the above issues, which upon applying the relevant changes has upgraded our manuscript!

Reviewer 2 Report (Previous Reviewer 1)

Comments and Suggestions for Authors

Dear Authors, 

I'm satisfied with your corrections.

Comments on the Quality of English Language

I have no comments.

Author Response

2nd reviewer

I have no comments.

Thanks!

Reviewer 3 Report (Previous Reviewer 3)

Comments and Suggestions for Authors

Thanks for your reply to my comments.

Now again it is difficult to revise all author´s changes since I am not sure to count with the last version and the line numbers in author´s comments do not correspond to the text in the web.

For example, two cases considering my original review and author´s reply:

My original review:

63 I do not understand the meaning and position here of the sentence “This was a

retrospective in-vitro/ex-vivo study”

Author´s reply:

We removed this sentence from the revised manuscript…. But this sentence is still there at line 65.

My original review:

I did not have access to proteome data (I think it was included in the revised manuscript).

Author´s reply

Supplementary figure 1 includes the proteome data…. but I do not have access now to supplementary figures.

Otherwise, I think the authors have made a decent work improving the manuscript.

Author Response

3rd reviewer

Thanks for your reply to my comments.

We thanks the reviewer for his comments and acknowledge his important contribution to the upgrading of the manuscript now.

Now again it is difficult to revise all author´s changes since I am not sure to count with the last version and the line numbers in author´s comments do not correspond to the text in the web.

We submit the latest version of the manuscript via the journal site. However, I will point out to the editorial office regarding this issue.

For example, two cases considering my original review and author´s reply:

My original review:

63 I do not understand the meaning and position here of the sentence “This was a

retrospective in-vitro/ex-vivo study”

Author´s reply:

We removed this sentence from the revised manuscript…. But this sentence is still there at line 65.

We apologize for that, it was removed now, thanks!

My original review:

I did not have access to proteome data (I think it was included in the revised manuscript).

Author´s reply

Supplementary figure 1 includes the proteome data…. but I do not have access now to supplementary figures.

   The vital access to the proteomic data which is presented in the supplementary figure 1 should be taken care of by the editorial office. I will specifically point it out to them in a separate mail.

Otherwise, I think the authors have made a decent work improving the manuscript.

Thanks so much again!!!

This manuscript is a resubmission of an earlier submission. The following is a list of the peer review reports and author responses from that submission.

Round 1

Reviewer 1 Report

Comments and Suggestions for Authors

The topic is very interesting but the quality of manuscript is poor.

In Section 2.

line 77 - What was the medium used?

line 88 - What was the concentration of antibiotics?

lines 84-85 - The sentence is unclear. In what volume and for how long the CLL cell were seeded?

lines 100, 144? 259 - Please, correct the Celsius presentation.

lines 104-105 - The sentence is unclear.

line 124 - 200 L of exosomes?

lines 291-293 - The sentence is unclear.

line 384 - What are these numbers?

Figure 1 - The images are too small and in poor quality. The caption for 1C is not describing what is presented on the figure. The numbers on scatterplots (1E) are unreadable. If you're talking about dose-dependant manner, please, present the data so that one could see the dependence, e.g. as a linear graph with linear regression coefficient.

Figure 3C - What is p=0.006 stands for?

Figure 4A - As for me, these images are not convincing. I don't see any dose dependency. Probably, the description is not clear, so I don't understand what exactly represent the dependency there.

Figure 5 - The quality of the picture is low, no word could be read.

lines 532-548 - Please, provide suitable references, not just "source" or "nih.gov".

Comments on the Quality of English Language

There are some frequent mistakes through the text. The number of the cells is presented wrongly (10 to the power of 3 is presented as 103, for example (line 128)).

The greek letters are frequently abscent or sometimes spelled in latin.

The citation in the text is wrongly represented through the text. Please, correct it.

line 136 - There are some extra parenthesis, please, correct. 

line 194 - The formula of water is written with mistake (2 should be in the lower index).

line 335 - Please, use commas between gene names.

Author Response

1st reviewer

The topic is very interesting but the quality of manuscript is poor.

In Section 2.

line 77 - What was the medium used?

We used exosome free media purchased from Biological-industries as outlined in the revised manuscript. 

line 78 - What was the concentration of antibiotics?

This information was added to the revised manuscript.

lines 84-85 - The sentence is unclear. In what volume and for how long the CLL cell were seeded?

Volume: 5ml, Time: 72h (see revised text).

lines 100, 144? 259 - Please, correct the Celsius presentation.

Done.

lines 104-105 - The sentence is unclear.

We rewrote this sentence as follows:

We stained these vesicles with anti-human CD19 conjugated to allophycocyanin (Milteniey) and with anti-CD81 conjugated to phycoerythrin (Milteniey) for 45 minutes.

line 124 - 200 L of exosomes?

This typo is corrected.

lines 291-293 - The sentence is unclear.

This is the sentence that appears instead in the revised manuscript: Since CD81 is ubiquitously expressed on all types of exosomes11 and CD19 is expressed by CLL cells and hence CLL exosomes, we doubled stained the particle extract with anti CD81 and anti CD19 (both purchased from Milteniey). To isolate the exosomes with used anti-CD81 magnetic beads diluted to 1:10 in PBS conjugated to phycoerythrin (Milteniey) for 45 minutes (Beckman) for 45 minutes.

line 384 - What are these numbers?

Sloppy editing, we thank the reviewer for noticing.

Figure 1 - The images are too small and in poor quality. The caption for 1C is not describing what is presented on the figure. The numbers on scatterplots (1E) are unreadable  If you're talking about dose-dependant manner, please, present the data so that one could see the dependence, e.g. as a linear graph with linear regression coefficien.  The size of figures are editorials' choice. For the sake of reviewing we asked to maintain the original size. Per this  reviewers' request we added dose- and time- dependent graphs (figure 1G)

Figure 3C - What is p=0.006 stands for?

Moved while editing, has been placed correctly in the revised manuscript.

Figure 4A - As for me, these images are not convincing. I don't see any dose dependency. Probably, the description is not clear, so I don't understand what exactly represent the dependency there.

Fig4B shows dose dependence increase in STAT3 phosphorylation. Figure 4A merely confirmed prior observations regarding IL-6 induced phosphorylation. In the revised manuscript the incorrect reference to figures was corrected and we thank the reviewer for noticing.

Figure 5 - The quality of the picture is low, no word could be read.

See revised manuscript for improved quality of the picture.

lines 532-548 - Please, provide suitable references, not just "source" or "nih.gov"

There are some frequent mistakes through the text. The number of the cells is presented wrongly (10 to the power of 3 is presented as 103, for example (line 128)).

These typos were corrected.

The Greek letters are frequently absent or sometimes spelled in Latin.

The citation in the text is wrongly represented through the text. Please, correct it.

line 136 - There are some extra parenthesis, please, correct. √

line 194 - The formula of water is written with mistake (2 should be in the lower index). √

line 335 - Please, use commas between gene names.

We omitted all unnecessary symbols and citations are now correctly presented  throughout. 

Reviewer 2 Report

Comments and Suggestions for Authors

The section on patients' characteristics lacks sufficient details. It would be beneficial to include information on the criteria used for selecting these specific patients for the study. Providing these details would enhance the readers' understanding of the patient population and the generalizability of the study findings.

 The author mentions using exosomes bound to anti-CD81 magnetic beads for flow cytometry analysis. It would be beneficial to explain the process of exosome isolation and the rationale behind choosing CD81 as the target marker. Additionally, the author should provide information on the source of the anti-CD81 magnetic beads and any relevant dilutions or incubation conditions used.

 Great work by the authors in demonstrating the uptake of CLL-exosomes by endothelial cells. The methodology used to isolate CLL-exosomes from patient PBMCs appears robust, with clear characterization using various techniques such as nanoparticle tracking analysis (NTA), electron microscopy (EM), and flow cytometry. The results confirm the presence of CD81-positive exosomes derived from CLL cells.

The authors further investigated the uptake of CLL-exosomes by HUVECs using confocal microscopy and flow cytometry. The use of membrane red dye FM-1-43 and FITC-conjugated peptide provided clear visualization of the stained exosomes inside the cells, validating the confocal microscopy data. The dose-dependent uptake of CLL-exosomes into HUVECs, as demonstrated by flow cytometry, highlights the importance of exosomal concentration in cellular internalization.

However, one aspect that requires clarification is how the authors determined the dose range of CLL-exosomes for the uptake experiments. Providing information on the rationale behind selecting the specific dose range, such as previous literature or pilot studies, would strengthen the experimental design and interpretation of the results.

 The study presents a very promising and connected story. However, one concern I have is regarding the quality and clarity of the gel images and the changes observed in the protein levels.

To ensure the robustness and reliability of the findings, it is essential that the gel images are of high quality and clearly demonstrate the changes in protein expression. Adequate resolution, appropriate exposure settings, and proper labeling of bands are crucial aspects to consider in gel image presentation.

Author Response

2nd reviewer

The section on patients' characteristics lacks sufficient details. It would be beneficial to include information on the criteria used for selecting these specific patients for the study. Providing these details would enhance the readers' understanding of the patient population and the generalizability of the study findings.

This information is added to the revised material and method section and to the revised table 1.  

 The author mentions using exosomes bound to anti-CD81 magnetic beads for flow cytometry analysis. It would be beneficial to explain the process of exosome isolation and the rationale behind choosing CD81 as the target marker. Additionally, the author should provide information on the source of the anti-CD81 magnetic beads and any relevant dilutions or incubation conditions used.

These details are added to the pargraph regarding exosome isolation in the revised manuscript.  We chose anti-CD81 since it is ubiquitiously expressed on exosomes and superior to other availbe magnetic beads for isolation purpose.

Great work by the authors in demonstrating the uptake of CLL-exosomes by endothelial cells. The methodology used to isolate CLL-exosomes from patient PBMCs appears robust, with clear characterization using various techniques such as nanoparticle tracking analysis (NTA), electron microscopy (EM), and flow cytometry. The results confirm the presence of CD81-positive exosomes derived from CLL cells.

   The authors further investigated the uptake of CLL-exosomes by HUVECs using confocal microscopy and flow cytometry. The use of membrane red dye FM-1-43 and FITC-conjugated peptide provided clear visualization of the stained exosomes inside the cells, validating the confocal microscopy data. The dose-dependent uptake of CLL-exosomes into HUVECs, as demonstrated by flow cytometry, highlights the importance of exosomal concentration in cellular internalization.

   However, one aspect that requires clarification is how the authors determined the dose range of CLL-exosomes for the uptake experiments. Providing information on the rationale behind selecting the specific dose range, such as previous literature or pilot studies, would strengthen the experimental design and interpretation of the results.

We chose the exosome range based on a dose- escalating experiments. To answer this quary we added figure 1G showing dose- and time- dependent uptake of the exosomes.  

The study presents a very promising and connected story. However, one concern I have is regarding the quality and clarity of the gel images and the changes observed in the protein levels.

   To ensure the robustness and reliability of the findings, it is essential that the gel images are of high quality and clearly demonstrate the changes in protein expression. Adequate resolution, appropriate exposure settings, and proper labeling of bands are crucial aspects to consider in gel image presentation.

Thank you for this comment. We carefully revised all Western Blot gel presented for resolution labeling of the bands and also revised the densitometry values.

Reviewer 3 Report

Comments and Suggestions for Authors

I have had access to your web site to the critical comments of the manuscript by three reviewers (and author´s reply). Nevertheless, the manuscript you provided me is the original version, and not the revised one. Thus, most of my comments have already been addressed by others and I even could not work on the modified manuscript.

Generally speaking, the manuscript looks “incomplete” in many aspects: quality of figures, formatting, spelling,… as stated by reviewers. Here are some comments not previously addressed.

Comments:

2. Material and Methods

63        I do not understand the meaning and position here of the sentence “This was a retrospective in-vitro/ex-vivo study

91        Authors isolate normal B cells from healthy donors as controls, but there is no reference to using them.

3. Results

As already mentioned by reviewers, numerous figures are too small and difficult to revise (Figure 1E, 1F,…).

318      3.1.1. Subsection”… Something is missing here?

315      Sometimes is used “b-catenin”, and others “betacatenin”

I did not have access to proteome data (I think it was included in the revised manuscript)

337      “…we selected 3 of the top upregulated phosphoproteins.” ¿Why this selection? Include information in the text referring to the level of these proteins against the other 53 upregulated ones.

Figure 2C values are not normalized.

Figure 3C. b-catenin protein expression does not look highly effective after transfection 353 vs 299 normalized values). ¿Have been quantified parasitic bands? ¿Are they b-catenin forms?

376- 383  This paragraph is difficult to follow to me. Please revise the text and the correspondence with the panels in Figure 4:

Figure 4A. There is no reference in the text (Figure 4A in the text should say 4B). There are 4 lanes per patient, and I do not have clear the content of each lane. Otherwise, this experiment refers to CLL cells incubated with “20 mg IL-6”, meanwhile next panel 4B, refers to CLL cells using HUVEC supernatants as an IL-6 source. ¿Why this discrepancy?

Figure 4B in the text, is actually, Figure 4C ???.

3. Discussion

I think it has been revised by the authors after previous comments.

Author Response

3rd reviewer

   I have had access to your web site to the critical comments of the manuscript by three reviewers (and author´s reply). Nevertheless, the manuscript you provided me is the original version, and not the revised one. Thus, most of my comments have already been addressed by others and I even could not work on the modified manuscript.

   Generally speaking, the manuscript looks “incomplete” in many aspects: quality of figures, formatting, spelling,… as stated by reviewers. Here are some comments not previously addressed.

comments:

Material and Methods

63        I do not understand the meaning and position here of the sentence “This was a retrospective in-vitro/ex-vivo study”

We removed this sentence from the revised manuscript.

91        Authors isolate normal B cells from healthy donors as controls, but there is no reference to using them.

We removed this sentence from the revised manuscript

Results

As already mentioned by reviewers, numerous figures are too small and difficult to revise (Figure 1E, 1F,…).

We scaled up figures that were too small to revise. With this we improved significantly the graphics of the manuscript.

318      “3.1.1. Subsection”… Something is missing here?

We removed this from the revised manuscript.

315      Sometimes is used “b-catenin”, and others “betacatenin”

In current version b-catenin is consistently used throught.

I did not have access to proteome data (I think it was included in the revised manuscript).

Supplementary figure 1 includes the proteome data.

337      “…we selected 3 of the top upregulated phosphoproteins.” ¿Why this selection? Include information in the text referring to the level of these proteins against the other 53 upregulated ones.

In the revised manuscript we provided the rationale for selecting these proteins and add comperative informations.  Briefly, bcatenin was selected based on the pathway analysis and 2 additional proteins that represent different cellular compartemnts (nuclear  and membranal).

Figure 2C values are not normalized.

Has been normalized in the revised manuscript thanks!!

Figure 3C. b-catenin protein expression does not look highly effective after transfection 353 vs 299 normalized values). ¿Have been quantified parasitic bands? ¿Are they b-catenin forms?

We agree and dedicated a lot of thought to this issue.  Since betacatenin is expressed in unmanipulated HUVECS as well as in other endothelial cells  we were skeptic as to whether a  moderate increase ( slightly less then 20%) will have any biological impact. Yet, our data confirms that this moderate overexpression resulted in consistent and reproducible results (see our  CHIP data).  In light of this reviewers' comment we added a commentary relating to this issue in the revised discussion.

376- 383  This paragraph is difficult to follow to me. Please revise the text and the correspondence with the panels in Figure 4:

To test whether “endothelial-induced IL-6” phosphorylates STAT3 we grew CLL cells in growth medium that was previously populated by endothelial cells that were exposed to CLL-exosomes. As controls, we grew CLL cells in the growth medium of unexposed endothelial cells.

Figure 4A. There is no reference in the text (Figure 4A in the text should say 4B). There are 4 lanes per patient, and I do not have clear the content of each lane. Otherwise, this experiment refers to CLL cells incubated with “20 mg IL-6”, meanwhile next panel 4B, refers to CLL cells using HUVEC supernatants as an IL-6 source. ¿Why this discrepancy?

Figure 4B in the text, is actually, Figure 4C

In response to the 3 above comments of this reviewer  (relating to the last paragraph of the results section) we revised the paragraph and the reference to the figurew and think that it is now more  comprihesible.

Discussion

I think it has been revised by the authors after previous comments.

We carfuly revised the discussion based on comments from previous and current reviewers suggestions.

Round 2

Reviewer 3 Report

Comments and Suggestions for Authors

No additional comments

Author Response

THANKS SO MUCH!!!!

:)